# CD40L Activates Platelet Integrin αIIbβ3 by Binding to the Allosteric Site (Site 2) in a KGD-Independent Manner and HIGM1 Mutations Are Clustered in the Integrin-Binding Sites of CD40L

**DOI:** 10.3390/cells12151977

**Published:** 2023-07-31

**Authors:** Yoko K. Takada, Michiko Shimoda, Yoshikazu Takada

**Affiliations:** 1Department of Dermatology, School of Medicine, UC Davis, Sacramento, CA 95817, USAmichiko.shimoda@ucsf.edu (M.S.); 2Department of Biochemistry and Molecular Medicine, School of Medicine, UC Davis, Sacramento, CA 95817, USA

**Keywords:** CD40L, integrins, αIIbβ3, High IgM syndrome type I (HIGM1), allosteric activation of integrins

## Abstract

CD40L is expressed in activated T cells, and it plays a major role in immune response and is a major therapeutic target for inflammation. High IgM syndrome type 1 (HIGM1) is a congenital functional defect in CD40L/CD40 signaling due to defective CD40L. CD40L is also stored in platelet granules and transported to the surface upon platelet activation. Platelet integrin αIIbβ3 is known to bind to fibrinogen and activation of αIIbβ3 is a key event that triggers platelet aggregation. Also, the KGD motif is critical for αIIbβ3 binding and the interaction stabilizes thrombus. Previous studies showed that CD40L binds to and activates integrins αvβ3 and α5β1 and that HIGM1 mutations are clustered in the integrin-binding sites. However, the specifics of CD40L binding to αIIbβ3 were unclear. Here, we show that CD40L binds to αIIbβ3 in a KGD-independent manner using CD40L that lacks the KGD motif. Two HIGM1 mutants, S128E/E129G and L155P, reduced the binding of CD40L to the classical ligand-binding site (site 1) of αIIbβ3, indicating that αIIbβ3 binds to the outer surface of CD40L trimer. Also, CD40L bound to the allosteric site (site 2) of αIIbβ3 and allosterically activated αIIbβ3 without inside-out signaling. Two HIMG1 mutants, K143T and G144E, on the surface of trimeric CD40L suppressed CD40L-induced αIIbβ3 activation. These findings suggest that CD40L binds to αIIbβ3 in a manner different from that of αvβ3 and α5β1 and induces αIIbβ3 activation. HIGM1 mutations are clustered in αIIbβ3 binding sites in CD40L and are predicted to suppress thrombus formation and immune responses through αIIbβ3.

## 1. Introduction

Integrins are a superfamily of cell-surface heterodimers that recognize extracellular matrix (ECM), cell-surface ligands (e.g., vascular cell adhesion molecule-1 (VCAM-1) and intercellular adhesion molecule-1 (ICAM-1)), and soluble molecules (e.g., growth factors). Integrins play important roles in biological processes (e.g., wound healing and hemostasis) and in the pathogenesis of diseases [1,2]. The CD40 ligand (CD40L) is a tumor necrosis factor family member, which is expressed in activated T cells and platelets as a transmembrane form and released as a soluble form (sCD40L) by proteolytic cleavage, on both T cells and platelets. CD40L is an inflammatory mediator and a major therapeutic target [3,4]. CD40L is stored in platelet granules and rapidly transferred to the surface upon platelet activation by platelet agonists (e.g., thrombin and ADP). CD40L in activated platelets has been shown to play a role in atherosclerosis formation and stabilizing thrombus [5,6].

CD40L binds to several integrins. Previous reports suggest that integrins αIIbβ3 [5], α5β1 [7], and αMβ2 [8] bind to CD40L. Each of these integrins interacts with CD40L in a specific manner. αIIbβ3 is known to recognize the KGD motif of CD40L (residues 115–117 of CD40L) [5]. α5β1 does not require the KGD motif [9]. Mutations of the CD40-binding site (Y145A, R203A, or Y145A/R203A double mutant) did not affect α5β1-CD40L interaction [10], suggesting that α5β1 and CD40-binding sites on CD40L are distinct. It has been reported that the N151A/Q166 mutation on the outside of trimeric CD40L suppresses α5β1-mediated CD40L binding [9]. Also, the functions of the α5β1–CD40L interaction are promoting tumor cell survival in the context of cancer and enhancing inflammation as well as T cell persistence in autoimmune conditions [3]. We recently showed that CD40L (residues 118–216) lacks the KGD motif bound to αvβ3, indicating that αvβ3 is a new CD40L receptor and CD40L binding to αvβ3 is not dependent on the KGD motif [11]. High IgM syndrome type 1 (HIGM1) is a congenital functional defect in CD40L/CD40 signaling due to defective CD40L [12]. HIGM1 is characterized by low or absent levels of serum IgG, IgA, and IgE, and normal or increased levels of serum IgM. These results from mutations in the pathway from B cell activation to isotype class switching due to CD40L mutations [12]. The CD40L-binding site in αvβ3 and α5β1 was located in the trimeric interface inside the CD40L trimer and, interestingly, HIGM1 mutants were clustered in the trimeric interface and these HIGM1 mutants are defective in integrin binding, indicating that functional defects in these HIGM1 mutants are related to defective integrin binding of CD40L. Also, CD40L bound to the classical ligand-binding site of αvβ3 (site 1) and αvβ3 site 1 recognized the binding site in the trimeric interface, in which several HIGM1 mutants were clustered [11]. These findings indicate that CD40L interaction with integrins αvβ3 and α5β1 is critical for immune responses and HIGM1 mutants affect integrin-CD40L interaction, leading to immune deficiency. Understanding the specifics of integrin-CD40L interaction, thus, is critical for new therapeutics for inflammatory diseases.

By virtual screening of a protein data bank using docking simulation, we identified the chemokine domain of CX3CL1 (fractalkine) [13] as a new ligand for integrins αvβ3, α4β1, and α5β1, and that CX3CL1 and integrins simultaneously bound to CX3CR1 [13]. The CX3CL1 mutant defective in integrin binding was defective in signaling and acted as an antagonist although it still bound to CX3CR1 [13], indicating that the direct integrin binding to CX3CL1 and subsequent integrin-CX3CL1-CX3CR1 ternary complex formation is required for its signaling functions. We unexpectedly discovered that CX3CL1 activated soluble integrin αvβ3 in 1 mM Ca^2+^ in cell-free conditions by binding to the allosteric ligand-binding site (site 2), which is distinct from site 1 in the integrin headpiece [14]. We assumed that the predicted allosteric site is open in the inactive form of integrin, in analogy to inactive allosteric enzymes, and we performed docking simulation of CX3CL1 and inactive (closed headpiece) integrin αvβ3 (1 JV2.pdb). CX3CL1 was shown to bind to the binding site distinct from site 1 [13]. Furthermore, we showed that peptides from site 2 of the integrin β subunit bound to CX3CL1 and blocked integrin activation by CX3CL1 [14]. Therefore, we concluded that site 2 is the allosteric binding site and involved in allosteric activation of integrins. We showed that CXCL12 (SDF-1, stromal cell-derived factor-1) activated integrins αvβ3, α4β1, and α5β1 by binding to site 2 [15], indicating that this mechanism of integrin activation is not limited to CX3CL1. We also found that CX3CL1, CXCL12, and CCL5 (Rantes) bound to site 1 of integrin αIIbβ3 and activated this integrin by binding to site 2, indicating that αIIbβ3 can be allosterically activated independent of the canonical inside-out signaling mechanism [16].

αIIbβ3 was identified as a receptor for CD40L [4,17] and is known to require the N-terminal KGD motif of CD40L (residues 115–117 of CD40L) for binding [4]. Using CD40L knockout (KO) mice and their wild-type (WT) littermates, CD40L was shown to be involved in thrombus stabilization, as evidenced by the increased thrombi rupture in injured arterioles of CD40L KO mice, owing to their low platelet content, compared with WT ones. Furthermore, treatment of CD40L KO platelets with recombinant sCD40L enhanced their thrombin-induced aggregation under high sheer stress conditions [5,6], as well as the capacity of sCD40L to induce tyrosine phosphorylation of the integrin β chain and the abrogation of such signals in the presence of antibodies directed against β3 tyrosine phosphorylation or in mice with mutated β3 tyrosine residues. By activating the integrin, CD40L promoted outside-in signaling in platelets, the formation of platelet microparticles, and enhancement of fibrinogen binding [17].

Activation of platelet integrin αIIbβ3 is a key event in platelet aggregation and thrombus formation. The specifics of CD40L binding to αIIbβ3 are unclear except that CD40L binding to αIIbβ3 suggestively requires the KGD motif [5]. We thus studied how CD40L binds to αIIbβ3 by using docking simulation and mutagenesis studies. For the present paper, we studied how CD40L binds to αIIbβ3 and whether CD40L activates αIIbβ3 in an allosteric manner. We showed that CD40L binds to αIIbβ3 in a KGD-independent manner using CD40L that lacks the KGD motif. Two HIGM1 mutants, S128E/E129G and L155, reduced the binding of CD40L to the classical ligand-binding site (site 1) of αIIbβ3, indicating that αIIbβ3 binds to the outer surface of CD40L trimer. Also, CD40L bound to site 2 of αIIbβ3 and allosterically activated αIIbβ3 without inside-out signaling. Two HIMG1 mutants K143T and G144E on the surface of trimeric CD40L suppressed CD40L-induced αIIbβ3 activation. These findings suggest that CD40L binds to αIIbβ3 in a manner different from that of αvβ3 and α5β1 and induces αIIbβ3 activation. HIGM1 mutations are clustered in αIIbβ3 binding sites in CD40L and are predicted to suppress thrombus formation and immune responses through αIIbβ3.

## 2. Materials and Method

### 2.1. Synthesis of Recombinant sCD40L with No KGD Motif

WT and mutant sCD40L [18] were synthesized as described [11]. sCD40L (residues 118–261) with N-terminal 6His tag (19 amino acid residues) migrated as a mostly single band with an expected molecular weight of monomer (163 amino acids, 18Kd) with a trace number of dimers and trimers. sCD40L mutants migrated as monomers with a size like that of WT CD40L (Appendix A).

### 2.2. Synthesis of Fibrinogen Fragments

Fibrinogen γ-chain C-terminal peptide. The [HHHHHH]NRLTIGEGQQHHLGGAKQAGDV (6 His tag and Fibrinogen γ-chain C-terminal residues 390–411) was conjugated with the C-terminus of GST (glutathione S-transferase) (designated γC390–411) in pGEXT2 vector (BamHI/EcoRI site) [19]. The protein was synthesized in E. coli BL21 and purified using glutathione affinity chromatography. The fibrinogen γ-chain C-terminal domain (γC151–411) was generated as previously described [20].

### 2.3. Binding of Soluble αIIbβ3 to CD40L

ELISA (enzyme-linked immunosorbent assay)-type binding assays were performed as described previously [13]. Briefly, wells of 96-well Immulon 2 microtiter plates (Dynatech Laboratories, Chantilly, VA, USA) were coated with 100 µL PBS containing sCD40L for 2 h at 37 °C. Remaining protein-binding sites were blocked by incubating with PBS/0.1% BSA for 30 min at room temperature. After washing with PBS, soluble recombinant αIIbβ3 (AgroBio, Almería, Spain, 1 µg/mL) was added to the wells and incubated in HEPES-Tyrodes buffer (10 mM HEPES, 150 mM NaCl, 12 mM NaHCO_3_, 0.4 mM NaH_2_PO_4_, 2.5 mM KCl, 0.1% glucose, 0.1% BSA) with 1 mM MnCl_2_ for 1 h at room temperature. After unbound αIIbβ3 was removed by rinsing the wells with binding buffer, bound αIIbβ3 was measured using anti-integrin β3 mAb (AV-10) followed by HRP (horse radish peroxidase)-conjugated goat anti-mouse IgG and peroxidase substrates.

### 2.4. Activation of Soluble αIIbβ3 by sCD40L

ELISA-type activation assays were performed as described previously [15]. Briefly, wells of 96-well Immulon 2 microtiter plates were coated with 100 µL PBS containing γC390–411 for 2 h at 37 °C. Remaining protein-binding sites were blocked by incubating with PBS/0.1% BSA for 30 min at room temperature. After washing with PBS, soluble recombinant αIIbβ3 (AgroBio, 1 µg/mL) was pre-incubated with sCD40L (WT or mutant) for 10 min at room temperature and was added to the wells and incubated in HEPES-Tyrodes buffer with 1 mM CaCl_2_ for 1 h at room temperature. After unbound αIIbβ3 was removed by rinsing the wells with binding buffer, bound αIIbβ3 was measured using anti-integrin β3 mAb (AV-10) followed by HRP-conjugated goat anti-mouse IgG and peroxidase substrates.

### 2.5. Activation of Cell-Surface αIIbβ3 on Chinese Hamster Ovary (CHO) Cells

αIIbβ3-CHO cells [21] were cultured in Dulbecco’s modified Eagle’s medium (DMEM)/10% fetal calf serum. The cells were resuspended with HEPES-Tyrodes buffer/0.02% BSA (heat treated at 80 °C for 20 min to remove contaminating cell adhesion molecules). The αIIbβ3-CHO cells were then incubated with WT or mutant sCD40L for 30 min on ice and then incubated with FITC-labeled integrin αIIbβ3 ligand (γC390–411) for 30 min at room temperature. The cells were washed with PBS/0.02% BSA and analyzed by BD Accuri flow cytometer (Becton Dickinson, Mountain View, CA, USA). The data were analyzed using FlowJo 7.6.5.

### 2.6. Docking Simulation

Docking simulation of interaction between CD40L (1ALY.pdb) and integrin αIIbβ3 was performed using AutoDock3, as described [22]. In the current study, we used the headpiece (residues 1–438 of αIIb and residues 55–432 of β3) of αIIbβ3 (5HDB.pdb). Cations were not present in αIIbβ3 during docking simulation [23,24]. The classical ligand-binding site (site 1) or the allosteric site (site 2) of αIIbβ3 were selected as the targets for CD40L.

### 2.7. Statistical Analysis

Treatment differences were tested using ANOVA and a Tukey multiple comparison test to control the global type I error using Prism 7 (GraphPad Software).

## 3. Results

### 3.1. sCD40L Binds to Soluble αIIbβ3 in a KGD-Independent Manner

Previous studies showed that CD40 binding to αIIbβ3 required the KGD motif (residues 215–217) of CD40L [4]. But the specifics of this interaction are unclear. The KGD motif is located at the N-terminus of soluble CD40L (sCD40L) (Figure 1a). To localize the αIIbβ3 binding site in CD40L, we performed docking simulation of interaction between the classical ligand-binding site (site 1) of αIIbβ3 (5hdb.pdb) and monomeric CD40L (1ALY.pdb) using Autodock3. The simulation predicted that αIIbβ3 binds to the outer surface of CD40L (docking energy −24.3 kcal/mol) and that the KGD motif is not in the integrin-binding interface of CD40L (Figure 1b). This is not consistent with the previous report that CD40L bound to αIIbβ3 in the KGD-motif-dependent manner [4].

We generated CD40L in which the KGD motif was deleted (residues 118–261) to study the role of the KGD motif (Appendix A). We studied whether the sCD40L (residues 118–261) binds to soluble αIIbβ3 in ELISA-type binding assays. Wells of a 96-well microtiter plate were coated with sCD40L, and remaining protein-binding sites were blocked with BSA. Wells were incubated with soluble αIIbβ3 for 1 h at room temperature in 1 mM Mn^2+^ (to fully activate integrin) and bound αIIbβ3 was quantified using anti-β3 and HRP-conjugated anti-mouse IgG. αIIbβ3 bound to the soluble CD40L (residues 118–261) in a dose-dependent manner (Figure 1c). This indicates that αIIbβ3 binding to CD40L does not require the KGD motif. A known ligand to αIIbβ3 (the disintegrin domain of ADAM15) [25,26] suppressed sCD40L binding to αIIbβ3 (Figure 1d), indicating that the binding of sCD40L to αIIbβ3 is specific.

### 3.2. Effect of a Group of HIGM1 Mutations on the Binding of sCD40L to the Classical Ligand-Binding Site (Site 1) of αIIbβ3

Positions of amino acid residues predicted to be involved in integrin binding are shown in Figure 2a,b. Notably, several amino acid residues mutated in HIGM1 (Ser128, Glu129, Lys143, and Leu155) are clustered in the integrin-binding interface (Figure 2a,b). We incubated soluble αIIbβ3 with CD40L HIGM1 mutants (S128E/E129R, K143T, and L155P) immobilized to wells for 1 hr at room temperature in 1 mM Mn^2+^. Bound soluble αIIbβ3 was quantified with anti-β3 mAb and HRP-conjugated anti-mouse IgG. Notably, we found that S128E/E129R and L155P mutations (but not K143T) suppressed the binding of soluble αIIbβ3 in 1 mM Mn^2+^ (Figure 2c). These findings suggest that amino acid residues of CD40L that are critical for site 1 binding are located outside of CD40L (Figure 2e), and that S128E/E129R and L155P HIGM1 mutants are defective in αIIbβ3 binding. A previous study reported that the binding site for αvβ3 and α5β1 in CD40L was mapped in the trimeric interface of CD40L (e.g., Y170, G226, and H224) [11]. This indicates that the αvβ3 and αIIbβ3 (site 1) bind to CD40L in distinct manners (Figure 2e) and is consistent with the prediction that the KGD motif is not involved in CD40L-αIIbβ3 interaction.

### 3.3. CD40L Activates Soluble and Cell-Surface Integrin αIIbβ3 in 1 mM Ca^2+^ by Binding to Site 2

Previous studies showed that chemokines CX3CL1, CXCL12, and CCL5 activated integrin αvβ3 and αIIbβ3 in 1 mM Ca^2+^, in which integrins are usually kept inactive, by binding to the allosteric site (site 2) [14,15,16]. The presence of site 2 was predicted to be on the opposite side of the classical ligand (RGD)-binding site (site 1) by docking simulation using the closed headpiece αvβ3 (1 JV2.pdb) as a target. We previously showed that CD40L can activate integrins αvβ3, α5β1, and α4β1 by binding to site 2 [18]. CD40L is rapidly transported to the surface of platelets upon platelet activation, but it is unclear if αIIbβ3 can be activated by CD40L by binding to site 2.

We used ELISA-type activation assays, in which soluble integrins were incubated with ligands that were immobilized to plastic in the presence of chemokines or other activators in 1 mM Ca^2+^ (to keep integrins inactive) in cell-free conditions [14]. Integrin activation was defined as the increase in the binding of soluble integrins to immobilized ligand. αIIbβ3 is known to bind to the C-terminal 400HHLGGAKQAGDV411 sequence of fibrinogen γ-chain C-terminal domain (γC) [27]. We used the C-terminal residues 390–411 of fibrinogen γ-chain fused to GST (designated γC390–411) [16] as a ligand for αIIbβ3. To study if CD40L activates αIIbβ3, soluble αIIbβ3 was pre-incubated with sCD40L for 10 min at room temperature and then incubated with immobilized γC390–411 for 1 hr at room temperature in the presence of sCD40L in 1 mM Ca^2+^. Bound soluble αIIbβ3 was quantified with anti-β3 mAb and HRP-conjugated anti-mouse IgG. Activation was defined as the increase in the binding of soluble αIIbβ3 to immobilized γC390–411. Notably, WT sCD40L activated αIIbβ3 in a dose-dependent manner in 1 mM Ca^2+^ (Figure 3a), indicating that sCD40L activates αIIbβ3 in cell-free conditions without inside-out signaling.

Binding of integrins to their ligands requires the presence of divalent cations, and different cations can strikingly alter integrin affinities to fibronectin [28]; Mn^2+^ produces the most striking increase in integrin ligand affinity compared with other divalent cations (Mg^2+^ or Ca^2+^) in a wide variety of integrins. Consequently, Mn^2+^ has been widely used as a positive control for integrin activation. Mn^2+^-recreated integrin activation was thought to mimic physiologic integrin activation because it activates integrins in the absence of a bound ligand and induces similar epitope exposure [29]. We used 1 mM Mn^2+^ as a standard for integrin activation. We found that sCD40L activated αIIbβ3 at a level comparable to that of 1 mM Mn^2+^ (Figure 3b), indicating that CD40L is a potent integrin αIIbβ3 activator. Although high concentrations of sCD40L were required to detect activation of soluble αIIbβ3, CD40L is a transmembrane protein and is expected to be highly concentrated on the surface.

CHO cells that express recombinant αIIbβ3 on the cell surface (αIIbβ3-CHO cells) were incubated with FITC-labeled γC390–411 in the presence of sCD40L and the binding of FITC-labeled γC390–411 was assayed in 1 mM Ca^2+^ in flow cytometry (Figure 3c). We found that sCD40L activated cell-surface αIIbβ3 (Figure 3d) in a dose-dependent manner.

### 3.4. A Group of HIGM1 Mutations Suppress Activation of αIIbβ3 by CD40L

We performed docking simulation of interaction between monomeric CD40L and αIIbβ3 (site 2) by targeting the allosteric ligand-binding site (site 2) of αIIbβ3 (Figure 4a). Amino acid residues involved in CD40L-αIIbβ3 site 2 interaction are shown in Figure 4b. The predicted site 2 binding site in CD40L is outside of the trimer and overlaps with that of the site 1 binding site (Figure 2a). Notably, 4 HIGM1 mutations are clustered in the site 2 binding interface (S128E/E129R, K143T, G144E, and L155P) of CD40L. We found that K143T and G144E were defective in activating soluble αIIbβ3 (Figure 4c,d), which is consistent with the prediction. The positions of K143T and G144E on CD40L are shown (Figure 4e), indicating that the site 1 and site 2 binding sites overlap on the outer surface of trimeric CD40L and two HIGM1 mutants were defective in activating αIIbβ3.

## 4. Discussion

Previously, our studies showed that direct binding of CD40L to integrin αvβ3 plays a critical role in CD40L signaling [11]. We showed that site 1 of αvβ3 bound to the trimeric interface of CD40L. It has been reported that some of the altered residues in HIGM1 are located in the protein core or trimeric interface (Y170C, Q174R, T176I, A208D, H224Y, G226A, G227V, and L258S), pointing to the roles of these HIGM1 mutations in structural instability or disruptive trimer formation leading to defective CD40L functions [30]. We showed that, in fact, these HIGM1 mutants are defective in integrin binding, and we proposed that the defective integrin binding caused functional defects in these HIGM1 mutations in immune responses as evidenced by reduced activation in B cells when stimulated with HIGM1 mutants compared to normal CD40L [11]. Also, our studies suggest that the αIIbβ3 (site 1)-binding site is distinct from the site 1 binding site of αvβ3 [11].

CD40L was shown to be involved in thrombus stabilization in addition to immune responses. αIIbβ3 is a key component of platelet aggregation and thrombus formation and binds to CD40L. Previous studies showed that αIIbβ3 binding to CD40L requires the KGD motif of CD40L (residues 115–117 of CD40L) [4,17]. We showed, however, that CD40L without KGD motif did bind to this αIIbβ3 integrin. This was supported by the docking model, which showed that the KGD motif is not close to the αIIbβ3-binding interface of CD40L. Instead, the αIIbβ3 (site 1)-binding site in CD40L is located outside of the CD40L trimer. Notably, a group of HIGM1 mutations (S128E/E129R and L155P), which was confirmed by mutagenesis studies with an in vitro binding assay, are clustered in the αIIbβ3 site 1 binding site, and suppressed CD40L binding to αIIbβ3 site 1. We previously reported that sCD40L bound to site 2 of αvβ3 and activated soluble αvβ3 [18]. Interestingly, the site 2 binding site in CD40L was in the region around the HIGM1 mutants S128R/E129G, K143T, G144E, and L155P, indicating that the αvβ3 site 2 binding site, and the αIIbβ3 site 1 and site 2 binding sites, overlap in CD40L. Notably, these mutants were defective in activating αvβ3 and acted as antagonists for CD40L-mediated αvβ3 activation [18]. These findings excluded the possibility that these CD40L mutants are not properly folded. Also, it would be interesting to study whether these CD40L mutants act as antagonists in CD40L-mediated αIIbβ3 activation and thrombus formation in future research.

Notably, in the present study, we showed that CD40L activated soluble integrin αIIbβ3 in cell-free conditions in 1 mM Ca^2+^. This was clear evidence that CD40L can activate αIIbβ3 by binding to site 2 without inside-out signaling. CD40L also activated cell-surface αIIbβ3 on CHO cells that did not have machinery for inside-out signaling. The αIIbβ3-binding site in CD40L is located outside of the CD40L trimer around two HIGM1 mutations, K143T and G144E, which is close to the recently mapped αvβ3 (site 2)-binding site [18]. Interestingly, this site overlaps with the αIIbβ3 (site 1)-binding site.

In sum, the present study suggests that the αIIbβ3 site 1 and site 2 binding sites, and the αvβ3 site 2 binding site in CD40L in the vicinity of the HIGM1 mutants S128R/E129G, K143T, G144E, and L155P, are critically involved in CD40L signaling functions. We propose that the defective immune responses in HIGM1 may be due to the defective integrin binding/activation, since HIGM1 mutations are clustered in the integrin-binding sites in αIIbβ3, αvβ3, and α5β1, as shown by our present and previous studies (Figure 5).

It has been established that, in general, integrin αIIbβ3 is activated by inside-out signaling upon the activation of platelets by thrombin and other platelet agonists. However, it has been proposed that inside-out signaling does not enhance ligand binding affinity to monovalent ligand [31,32]. Inside-out signaling induces integrin binding to multivalent ligands (e.g., ligand-mimetic antibody, Pac-1 IgM specific for αIIbβ3, with potential 10-binding sites) [33,34]. Since the site 2-mediated integrin activation enhanced binding affinity to monovalent ligands, we propose that site 2-mediated integrin activation may be a potential mechanism for enhancing the ligand affinity of integrins. We have recently shown that several pro-inflammatory factors, CX3CL1, CCL5, and CXCL12, activate soluble integrins αIIbβ3 and αvβ3 by binding to the allosteric site (site 2) [16], and CD40L in the present study. We recently showed that CD62P (P-selectin) binds to αIIbβ3 and αvβ3 and activates them by binding to site 2 [19]. In this context, it would be interesting to study whether other cytokines and chemokines stored in platelet granules may also interact with αIIbβ3 and concomitantly amplify and stabilize the process of platelet activation. Such a mechanism might facilitate platelets having a multifaceted role as the amplifier of the immune responses through hemostasis and inflammatory cytokine release for recruiting immune cells such as monocytes and T cells.

CD40L is expressed on activated T cells (Figure 5). Our current findings have implications for the immune modification by platelets through direct interaction between platelets and immune cells. For example, activated platelets trigger secretion of cytokines and chemokines from monocytes and neutrophils through forming heterotypic cell aggregates [35]. Also, platelets are known to form complexes with lymphocytes [36] and fine tune their differentiation, including CD4 T helper and Treg differentiation [37,38,39,40], anti-viral cytotoxic CD8 T cell response [41], and B cell antibody production [42,43]. Importantly, platelets display major histocompatibility complex (MHC) and directly stimulate T cells [44]. Such platelet and lymphocyte interactions are thought to play an essential role in cardiovascular diseases [45] and COVID-19 [46,47]. However, the detailed mechanisms by which platelets form stable complexes with lymphocytes are not entirely clear. Notably, activated monocytes and B cells both express CD40, the receptor of CD40L. Therefore, based on our previous and current findings, and given the fact that CD40L can bind and activate other integrins such as α5β1 and α4β1 on lymphocytes or αvβ3 on vascular endothelial cells, we hypothesize that direct CD40L and integrin interaction plays a critical role in the interface of platelet and immune cell complexes for immune cell activation. Further studies will be needed to fully understand the specifics of CD40L-integrins interaction and immune cell activation in inflammatory settings, which can form the basis to develop new therapeutics for inflammatory diseases.

## Figures and Tables

**Figure 1 cells-12-01977-f001:**
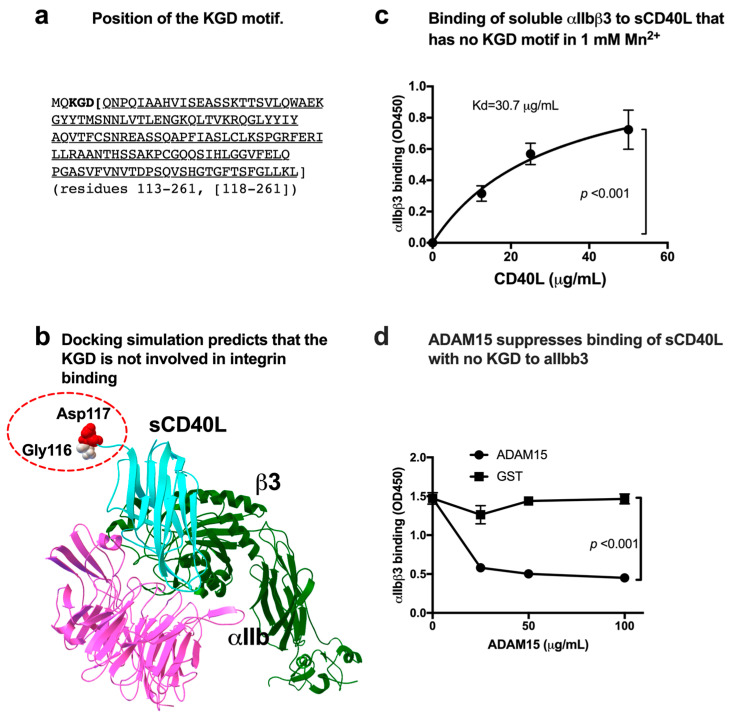
Binding of soluble CD40L to αIIbβ3 does not require the KGD motif. (**a**) Amino acid sequence of sCD40L (118–261) that lacks the KGD motif. We synthesized CD40L (118–261) to test whether the KGD motif is critical for αIIbβ3 binding. (**b**) Docking simulation of sCD40L and integrin αIIbβ3. Docking simulation was performed as described in the method section. Monomeric sCD40L was predicted to bind to the classical ligand-binding site of αIIbβ3 with docking energy −24.3 kcal/mol. (**c**) Binding of soluble integrin αIIbβ3 to sCD40L that lacks the KGD motif in 1 mM Mn2+. Wells of 96-well microtiter plate were coated with sCD40L. Remaining protein-binding sites were blocked with BSA. Wells were incubated with soluble αIIbβ3 (1 μg/mL) in Tyrode-HEPES buffer with 1 mM Mn2+ (to activate αIIbβ3) for 1 h at room temperature. After removing the unbound integrin by washing, bound αIIbβ3 was quantified using anti-β3 mAb (AV10) and HRP-conjugated anti-mouse IgG. Data are shown as means +/− SD in triplicate experiments. (**d**) The disintegrin domain of ADAM15 (A disintegrin and metalloprotease-15) inhibits the binding of αIIbβ3 to immobilized sCD40L (118–261). The binding of soluble αIIbβ3 to CD40L was performed as described in (**c**), except that the disintegrin domain of ADAM15 fused to GST or parent GST was added. Data are shown as means +/− SD in triplicate experiments.

**Figure 2 cells-12-01977-f002:**
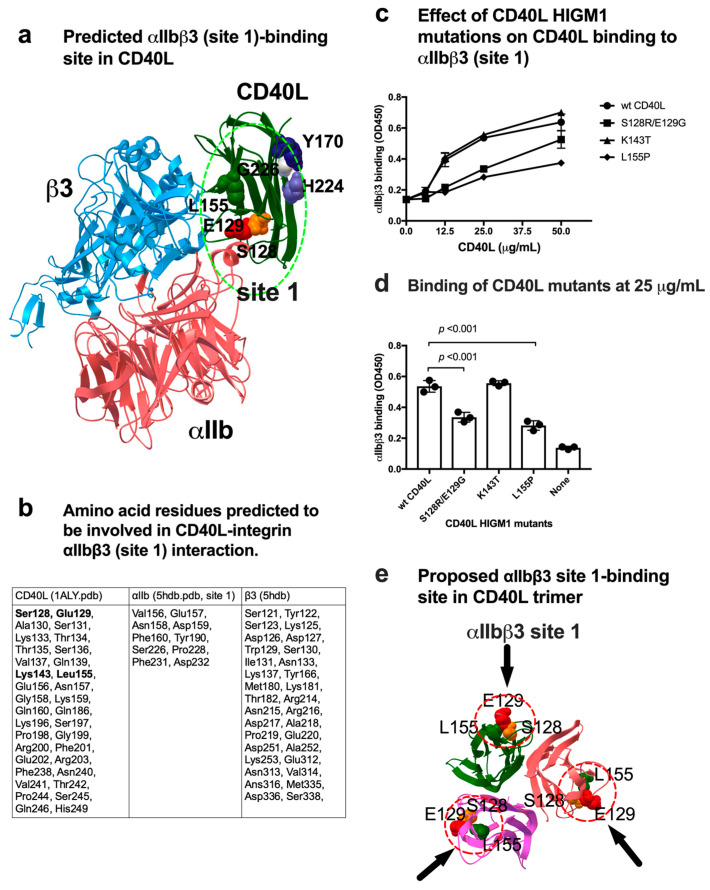
Effect of HIGM1 mutations on αIIbβ3 binding to CD40L. (**a**) Positions of HIGM1 mutations (S128, E129, and L155) in the predicted binding interface between αIIbβ3 (site 1) and CD40L monomer. Y170 and H224 are in the trimeric interface and are involved in binding to αvβ3 site 1 [11]. (**b**) Amino acid residues of CD40L that are predicted to be involved in binding to αIIbβ3. Amino acid residues within 0.6 nm between CD40L and αIIbβ3 in the docking model were selected using pdb viewer (version 4.1). Amino acid residues in CD40L that are mutated in HIGM1 are shown in bold. (**c**) Effect of the HIGM1 mutations on the binding of soluble αIIbβ3 to sCD40L. We selected several HIGM1 mutations in the site 1 binding interface of CD40L predicted by the simulation. The binding of CD40L mutants to soluble αIIbβ3 was studied as described in Figure 1. The binding of CD40L mutants at 25 μg/mL coating condition is shown. Data are shown as means +/− SD in triplicate experiments. (**d**) Positions of S128, E129, and L155P that affect the binding of CD40L trimer to soluble αIIbβ3. (**e**) Proposed αIIbβ3 site 1-binding site in CD40L trimer.

**Figure 3 cells-12-01977-f003:**
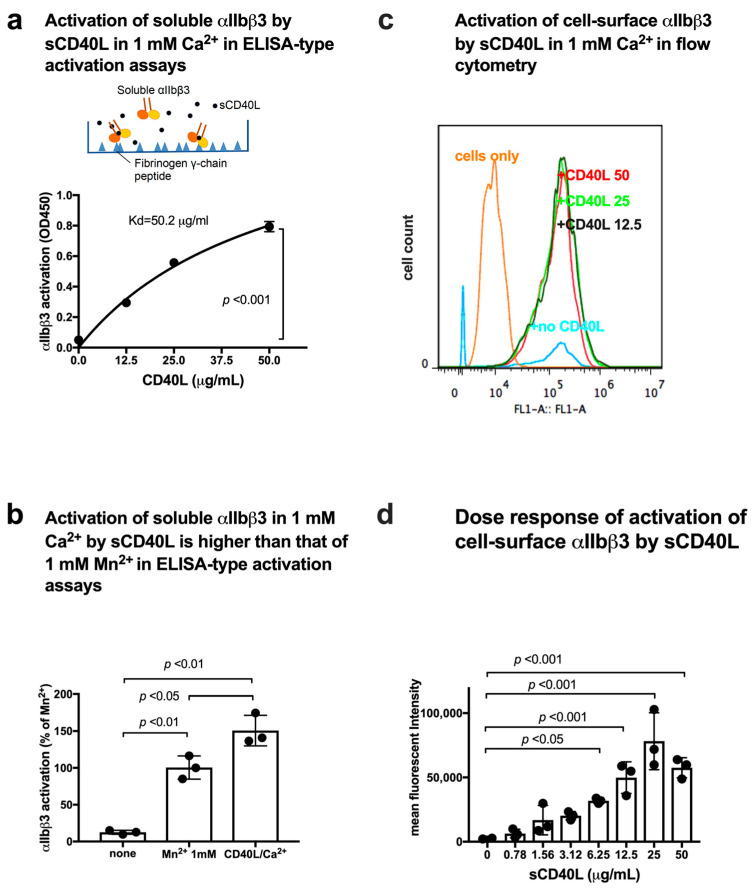
sCD40L activates soluble αIIbβ3 in 1 mM Ca^2+^. (**a**) CD40L activated αIIbβ3 in a dose-dependent manner. Wells of 96-well microtiter plate were coated with γC390–411 (the αIIbβ3-ligand peptide conjugated to GST) (20 µg/mL) and the remaining protein-binding sites were blocked with BSA. Wells were incubated with soluble αIIbβ3 (1 µg/mL) in Tyrode-HEPES buffer with 1 mM Ca^2+^ (to keep αIIbβ3 inactive) for 1 h at room temperature in the presence of 50 µg/mL CD40L. After washing the unbound integrin, bound αIIbβ3 was quantified using anti-β3 mAb (AV10) and HRP-conjugated anti-mouse IgG. Data are shown as mean +/− SD in triplicate experiments. (**b**) Activation of soluble αIIbβ3 in 1 mM Ca^2+^ by sCD40L was higher than that of 1 mM Mn^2+^ in ELISA-type activation assays. Activation of soluble αIIbβ3 by sCD40L was determined as described in (**a**). 1 mM Mn^2+^ was used instead of 1 mM Ca^2+^ as a positive control (as 100%). Data are shown as mean +/− SD in triplicate experiments. (**c**) Activation of cell-surface αIIbβ3 on CHO cells in flow cytometry. αIIbβ3-CHO cells were incubated with WT sCD40L (0–50 µg/mL) for 30 min on ice and then incubated with FITC-labeled integrin αIIbβ3 ligand γC390–411 for 30 min at room temperature. The cells were washed with PBS/0.02% BSA and analyzed by BD Accuri flow cytometer (Becton Dickinson, Mountain View, CA, USA). The data were analyzed using FlowJo 7.6.5. (**d**) Dose response of activation of cell-surface αIIbβ3 by soluble CD40L. Activation of cell-surface αIIbβ3 was determined as described in (**c**). Data are shown as mean +/− SD in triplicate experiments. Values with no sCD40L were subtracted.

**Figure 4 cells-12-01977-f004:**
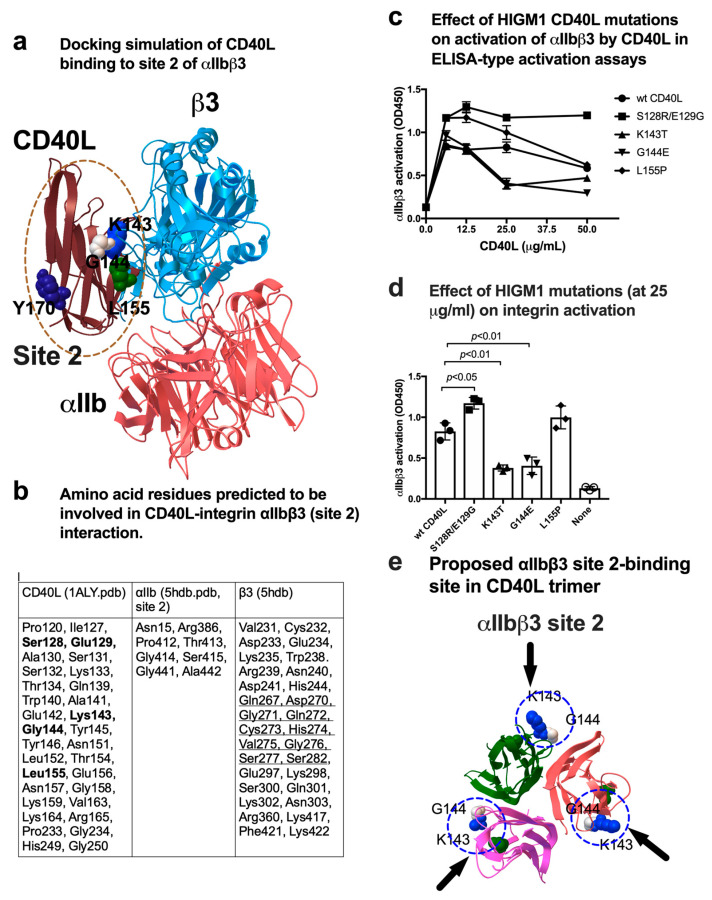
Effect of HIGM1 mutations on activation of αIIbβ3 by CD40L. (**a**) Positions of HIGM1 mutations in the CD40L-integrin αIIbβ3 (site 2) complex model. Y170 is in the trimeric interface of CD40L trimer. (**b**) Amino acid residues within 0.6 nm between CD40L and αIIbβ3 were selected using pdb viewer (version 4.1). Amino acid residues in CD40L that are mutated in HIGM1 are shown in bold. Amino acid residues in integrin β3 that are involved in site 2 peptide are underlined. (**c**) Effect of HIGM1 mutations on the CD40L-mediated activation of αIIbβ3. We studied the effect of several HIGM1 mutations on the CD40L-mediated activation of soluble αIIbβ3. Wells were coated with γC390–411 and incubated with soluble αIIbβ3 (1 µg/mL) in Tyrode-HEPES buffer with 1 mM Ca^2+^ (to keep αIIbβ3 inactive) for 1 h at room temperature in the presence of CD40L (wt and HIGM1 mutants). After washing the unbound integrin, bound αIIbβ3 was quantified using anti-β3 mAb (AV10) and HRP-conjugated anti-mouse IgG. Data are shown as mean +/− SD in triplicate experiments. (**d**) Effect of HIGM1 mutations (at 25 µg/mL) on integrin activation. (**e**) The HIGM1 mutations that affected activation of soluble αIIbβ3 are located outside of CD40L trimer.

**Figure 5 cells-12-01977-f005:**
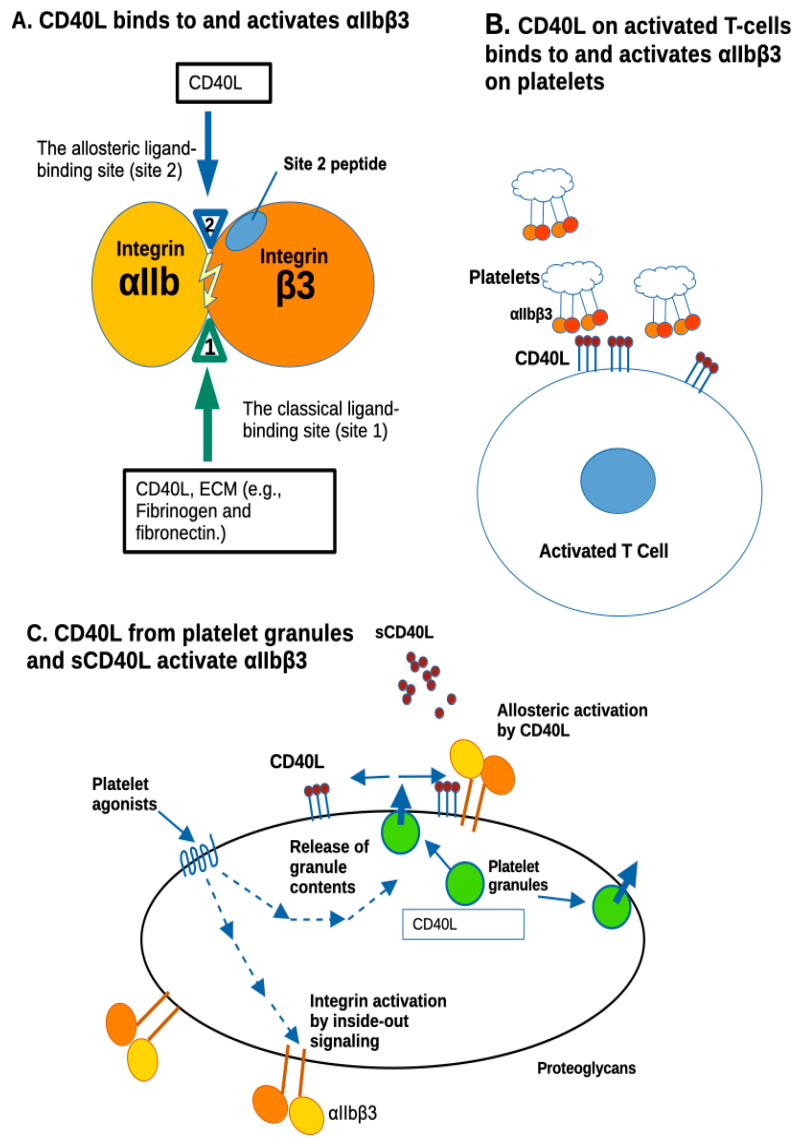
Allosteric activation of αIIbβ3 by CD40L. (**A**) The present study showed that CD40L binding to αIIbβ3 (site 1) does not require the KGD motif (at the N-terminus of sCD40L). CD40L also binds to site 2 and activates αIIbβ3. CD40L binding interfaces to αIIbβ3 (site 1 and site 2) are exposed to the surface of trimeric CD40L and overlap. Notably, HIGM1 mutations are clustered in this area. Peptide from site 2 was shown to bind to sCD40L [18], indicating that CD40L binds to site 2. CD40L binding interface to site 1 of integrins αvβ3 and α5β1 are in the trimeric interface (cryptic in CD40L trimer), indicating that CD40L monomer can bind to these integrins, but not trimer [11]. CD40L binds to site 2 and activates αvβ3 and α5β1. The binding interface to site 2 of αvβ3 and α5β1 overlap with that of αIIbβ3 [18]. Notably, HIGM1 mutations are clustered in the site 1 and site 2 binding interfaces, indicating that immune deficiency of HIGM1 may be due to the defect in integrin binding and/or activation. (**B**) The present study predicted that CD40L on activated T cell surfaces can bind to mediate T-cell–platelet interaction by binding to and activating platelet integrin αIIbβ3. Although αIIbβ3 on inactive platelets is not activated, CD40L on activated T cells may activate it by binding to site 2 of αIIbβ3. (**C**) It is known that CD40L is stored in platelet granules and rapidly transported to the platelet surface upon platelet activation. CD40L is predicted to activate αIIbβ3 by binding to site 2 of αIIbβ3 on the surface upon platelet activation. Soluble CD40L levels increase during inflammation and may induce allosteric activation of αIIbβ3 and platelet aggregation.

## Data Availability

Research data is available upon request.

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
