# Peer review of "CD40L Activates Platelet Integrin αIIbβ3 by Binding to the Allosteric Site (Site 2) in a KGD-Independent Manner and HIGM1 Mutations Are Clustered in the Integrin-Binding Sites of CD40L"

_cells, 2023, doi:10.3390/cells12151977_

Round 1
Reviewer 1 Report
This is an elegantly done study with nice figures.
This paper would be strengthened by adding a summative conclusion statement at the end of the discussion as well as addressing potential limitations.
Detailed language editing should be undertaken. I noticed several instances of tense, parts of speech, and plurality errors that should be corrected for a smoother read of this complex paper.
Author Response
Thank you for your kind comments.
We corrected grammatical errors. We already included summary of our findings in the discussion section, and Fig. 5 summarizes our findings. So we did not add the requested summary at the end.
Reviewer 2 Report
In the present work, the authors observed that CD40L binds and activates αIIbβ3, in a KGD independent manner, using CD40L that lacks the KGD motif. They also observed that HIGM1 mutations are clustered in αIIbβ3 binding sites in CD40L, which can predict the suppression of thrombus formation and immune responses through αIIbβ3 in patients with this syndrome. Please, see the criticisms and suggestions as follow.
1) Title is a little confusing.
2) Abbreviations. All abbreviations have to be defined at the first time.
3) Authors use several molecular strategies to show how CD40L binds and activates αIIbβ3 and the influence of HIGM1 mutations in this process. However, it is lacking a biological or physiological relevance of the study. Any downstream signaling pathway or any biological response was evaluated in response to the CD40L-αIIbβ3 interaction.
4) Based on the methodologies used, a description of the bias and limitations of the study need to be presented. Perspectives of the study must be presented in order to highlight the importance of the findings and to direct further studies.
5) English revision is required.
English revision is required.
Author Response
1) Title was changed. "CD40L activates platelet integrin αIIbβ3 by binding to the allosteric site (site 2) in a KGD-independent manner and HIGM1 mutations are clustered in the integrin-binding sites of CD40L"
2) Abbreviations were defined when they first appeared.
3 and 4) We focused on CD40L binding to integrin, and did not measure downstream signaling in this manuscript. We previously showed that HIGM1 CD40L mutants are antagonistic in avb3, and we will be able to show antagonistic effects in aIIbb3 in future studies. We added the following sentence (line 346)
"Also, it would be interesting to study if these CD40L mutants act as antagonists in CD40L-mediated αIIbβ3 activation and thrombus formation in future studies."
5) Grammatical errors were corrected.
Reviewer 3 Report
The study entitled 'CD40L allosterically activates platelet integrin αIIbβ3 and HIGM1 mutations are clustered in the integrin-binding sites of CD40L' mainly studied the binding sites of CD40L to integrin αIIbβ3 through prediction and experimental binding tests, and found that CD40L allosterically activates integrin αIIbβ3 through several binding sites to site 1 and site 2 on αIIbβ3.
Below is a few questions for the authors:
1. The current introduction is mainly about CD40L and integrin. There is no introduction about HIGM1. What are the symptoms of this disease? There should be at least a short description about it in the introduction.
2. What are GST and ADAM15 in Figure 1? Each abbreviation should have their full name stated the first time they appear. It is the same for vcam-1, ICAM-1, etc.
3. Sometimes the fonts in the text are not consistent which should be corrected. For example, the font in Figure 4's legend.
4. There are many amino acid residues predicted to be invovled in CD40L-aIIbb3 interaction. The authors choose a few amino acid residues to study, maybe because these residues have been found in the HIGM1 mutation. However, there is no introduction about the discoveries related to HIGM1 mutation, and it is not clear whether the HIGM1 mutation only includes these studied amino acid residues, or also include some other residues?
5. In Figure 4c, when the concentration of CD40L is lower than 12.5ug/ml, all groups have similar trend, and the difference between groups is not significant. And why is the trend of wt CD40L in this figure different from that of wt CD40L in Figure2C? These two are both wt CD40L and should remain consistent.
6. There is an error in the legend of Figure 4c. It says '...for 1 h at room temperature in the presence of 50 μg/ml CD40L'. But the actual test is 0-50 μg/ml CD40L. The same problem occurs in the legend of Figure2c, which says '... The binding of CD40L mutants at 25 μg/ml coating condition is shown', but the test is also 0-50 μg/ml CD40L.
7. Figure 3c:The no CD40L group has two peaks. The expression of αIIbβ3 in the second peak was actually the same with that in the CD40L treated group. The difference between the second peak of no CD40L group and the CD40L treatment group is only in that CD40 treatment eliminated the first peak.
8. The title of the article seems a bit strange, it is constituted by two separate sentences that should be combined into one sentence.
9. It is interesting to see that the binding sites on CD40L to site 1 and site 2 of αIIbβ3 are both those two amino acids (L155P, S128R), so here comes a practical question: when adding CD40L to αIIbβ3, which of the two sites has a higher binding force and which has a lower binding force? Which site on αIIbβ3 does CD40L preferentially bind to?
10. Another problem is that the experimental data says that the binding sites on CD40L to site 2 of αIIbβ3 are L155P and S128R, however the abstract states ' Two HIMG1 mutants K143T and G144E on the surface of trimeric CD40L suppressed CD40L-induced αIIbβ3 activation', which seems to say that the active site on CD40L to site 2 of αIIbβ3 are K143T and G144E?
Author Response
1) We added "HIGM1 is characterized by low or absent levels of serum IgG, IgA, IgE and normal or increased levels of serum IgM. They are resulting from mutations in the pathway from B-cell activation to isotype class switching due to CD40L mutations [14]. " (line 57)
2) Abbreviations were defined when they appear first.
3) Font in Fig. 4 was corrected.
4) HIGM1 mutations in CD40L are listed in the unirpot or other database. We previously found that many HIGM1 mutations are clustered in the integrin-binding interface of CD40L (ref 1 and 2) and found that these mutations affect integrin binding and allosteric activation of integrins. We previously mutated basic amino acid residues in the integrin-binding interface to Glu (charge-reversal mutations), which also worked. We realized that testing HIGM1 mutations is sufficient, because they affect integrin binding or activation almost without exception.
5) Fig. 2c and 4c are different. Fig. 2c shows binding to site 1 in 1 mM Mn2+. Fig. 4c shows activation of integrin by binding to site 2 in 1 mM Ca2+. Since the CD40L mutations are clustered in the integrin-binding sites of CD40L, it is not surprising that they affect integrin activation and binding to some extent. The concentration (25 ug/ml) is the optimum concentration to detect the difference among HIGM1 mutants.
6) Corrected. Thanks.
7) Most likely, the second peak of aIIbb3 without CD40L is constitutively activated aIIbb3. We speculate that small numbers of aIIbb3 is constitutively activated in the absence of CD40L (or other allosteric activatiors). avb3 is not activated without CD40L. However, we do not have data to prove this at this point.
8) Title was modified as requested.
9) In our model, site 2 is open and site 1 is not open in inactive aIIbb3 on non-activated platelets. Thus, it is expected that CD40L binds to site 2 first and activate this integrin (site 1 becomes open).
10) Our results show that K143T and G144E bind to site 2.
Reviewer 4 Report
The authors investigated the role of CD40L and the effects of its interactions with various integrins.
The study is very robust. The materials-methods and results are detailed. The discussion and conclusions are very interesting and provide the basis for further study.
Author Response
Thanks for encouraging comments.
Reviewer 5 Report
In this manuscript, the authors explored the binding mechanism of CD40L to αIIbβ3 through a combination of docking simulations and mutagenesis studies. The findings of the study reveal that CD40L interacts with αIIbβ3 in a KGD independent manner using CD40L without KGD motif. The article is well-written and provides well-illustrated results.
Minor revision is suggested.
1. In line 178, the author claimed that a previous report suggested CD40L binds to αIIbβ3 in a KGD motif-dependent manner, but the mentioned article does not provide this conclusion. Moreover, there are other published articles that indicate CD40L binds to αIIbβ3 in a KGD motif-independent manner. Could the author please provide an explanation for citing this article?
2. In line 258, the author stated that “sCD40L activates αIIbβ3 in cell-free conditions without inside-out signaling.” Could the author please provide further details regarding the relationship between the binding activity of CD40L and inside-out signaling
Author Response
1) Reference 7 was replaced.
2) CD40L activated soluble integrin in cell-free conditions and activated cell surface integrin on CHO cells, which are known to lack intracellular machineries for inside-out signaling. This was already included in the original version.
Round 2
Reviewer 3 Report
1. Ok
2. Thank you, I checked the manuscript, some abbreviations have now been defined when they first appear, but some others are not, such as CX3CL1,CX3CR1, CXCL12, CCL5, ELISA, etc.
3. Ok
4. Thank you for the clarification of why study HIGM1 mutations, but the reply has not answered my question, which is whether the HIGM1 mutations only include the studied amino acid residues, or whether there are some other amino acid mutant residues in the HIGM1 mutations that have not been studied?
5. Ok
6. Ok
7. Thank you, and wish your speculation can be proved in further studies.
8. Ok
9. Thank you
10. Ok
Author Response
Thank you for your comments.
#2. I included additional definitions for abbreviations (CX3CL1, CXCL12, CCL5, ELISA). CXCL1 etc is a standard nomenclature for chemokines. So, I included additional names (e.g., fractalkine etc).
#4. Many misssense CD40L HIGM1 mutations were listed in the unirpot database. We have systematically studied all these mutations. We described only mutations in the predicted CD40L-aIIbb3 interface in this paper. Many other HIGM1 mutations are not in the CD40L-aIIbb3 binding interface and were not studied in this paper. Many other mutations are in the predicted CD40L-avb3 or a5b1 interface (in the trimeric interface of CD40L) and were previously extensively studied ((1) Takada, Y.K., J. Yu, M. Shimoda, and Y. Takada, Integrin Binding to the Trimeric Interface of CD40L Plays a Critical Role in CD40/CD40L Signaling. J Immunol, 2019. 203(5): p. 1383-1391. 10.4049/jimmunol.1801630; (2) Takada, Y.K., M. Shimoda, E. Maverakis, B.H. Felding, R.H. Cheng, and Y. Takada, Soluble CD40L activates soluble and cell-surface integrins alphavbeta3, alpha5beta1 and alpha4beta1 by binding to the allosteric ligand-binding site (site 2). J Biol Chem, 2021: p. 100399. 10.1016/j.jbc.). We found most HIGM1 mutations affected integrin-CD40L interactions.